# Frequency, Distribution, and Antimicrobial Resistance of Methicillin-Resistant Staphylococci and *Mammaliicoccus sciuri* Isolated from Dogs and Their Owners in Rio de Janeiro

**DOI:** 10.3390/antibiotics14040409

**Published:** 2025-04-16

**Authors:** Fernanda Cruz Bonnard, Luciana Guimarães, Izabel Mello Teixeira, Sandryelle Mercês Freire, Alessandra Maia, Patrícia Câmara de Castro Abreu Pinto, Thais Veiga Blanchart, Bruno Penna

**Affiliations:** Laboratório de Cocos Gram Positivos, Instituto Biomédico, Universidade Federal Fluminense (UFF), Niterói 24210-130, Brazil; fernandabonnard@id.uff.br (F.C.B.); luguimaraesvet@gmail.com (L.G.); belmello93@gmail.com (I.M.T.); sandryellemerces@id.uff.br (S.M.F.); alessandramaia@id.uff.br (A.M.); patriciaccap@id.uff.br (P.C.d.C.A.P.); thaisblanchart@id.uff.br (T.V.B.)

**Keywords:** zoonosis, transmission, antimicrobial resistance, one health, public health

## Abstract

**Background:** *Staphylococcus* spp. and *Mammaliicoccus sciuri* (*M. sciuri*) are Gram-positive cocci that inhabit mammals’ and birds’ skin and mucous membranes, part of the microbiota. An imbalance in local immunity can increase colonization, resulting in various infections. Inappropriate use of antimicrobials leads to Staphylococci and *M. sciuri* becoming resistant to conventional treatments. The transmission of methicillin-resistant staphylococci and *M. sciuri* (MRS and MRMs, respectively) between humans and animals is still underreported in Brazil. **Objectives:** this study aimed to describe the frequency, distribution, resistance pattern, and evaluation of potential sharing of MRS and MRMs in isolates from asymptomatic dogs and their owners in Rio de Janeiro. **Methods:** Samples from 50 asymptomatic dogs and 34 from their owners were collected. Isolates were identified by mass spectrometry. The *mecA* gene was confirmed by conventional PCR. Antimicrobial activity of samples that carried the *mecA* gene was evaluated by disk diffusion method. **Results:** In this study, MRS and MRMs were analyzed in 50 dogs and their owners (34) by identifying strains carrying the *mecA* gene. A total of 185 strains were isolated. The *mecA* gene was found in 33.5% of the isolates. The most prevalent species carrying the *mecA* gene was *S. epidermidis* (33.9%). MRMs showed 14.5%. Fourteen dogs had the same strain carrying the *mecA* gene as their owners. Of these, 50% exhibited the same antimicrobial resistance pattern, determined by the disk diffusion. The highest percentage of resistance observed in the MRS isolated from dogs and the owners was to Erythromycin (51.3% and 56.5%, respectively). **Conclusions:** The presence of methicillin-resistant staphylococci is worrisome because there is the potential to transfer these strains between dogs and humans. These strains may act as a reservoir of resistance genes.

## 1. Introduction

Staphylococci are part of the normal microbiota of the skin and mucous membranes of humans and animals [1]. Up to now, 89 species and 30 subspecies have been described for the genus [2,3]. Thus, species of the genus are divided into Coagulase-Positive *Staphylococcus* (CPS) and Coagulase-Negative *Staphylococcus* (CNS) [4,5]. Among the CPS species, the most well-known are *S. aureus* and *S. pseudintermedius* [6].

The species most associated with human diseases is *Staphylococcus aureus* [4]. Although human colonization and infection by *S. aureus* are at the forefront on a global scale, the organism has a long evolutionary history as a multi-host opportunistic pathogen [7]. This species can cause various infections, from skin lesions to pneumonia, meningitis, endocarditis, toxic shock syndrome, and septicemia [8]. Methicillin-resistant *S. aureus* (MRSA) has emerged as a significant pathogen in hospitals, communities, and veterinary environments, compromising public health and livestock production. Additionally, the high pathogenicity of MRSA, due to a series of virulence factors produced by *S. aureus*, combined with antibiotic resistance, helps to breach host immunity and is responsible for causing severe infections in humans and animals [9]. MRSA strains have been considered a global public health problem.

*Staphylococcus pseudintermedius* is a significant component of the cutaneous microbiome of healthy dogs. It is also the leading cause of canine skin, ear, urinary tract infections, and post-surgical wounds. Methicillin-resistant *S. pseudintermedius* (MRSP) has also been reported as an emerging problem in veterinary medicine. These microorganisms are highly relevant in the clinical practice of small animals, as they are resident skin bacteria in dogs and are responsible for opportunistic infections [10]. Although not typically isolated from humans, cross-species transmissions have already been documented. Nonetheless, this zoonotic potential of *S. pseudintermedius* strains cannot be neglected, since these samples may be misidentified as *S. aureus*. Close contact between pets and their owners can facilitate zoonotic transmission [11].

First classified as an occasional pathogen, *M. sciuri*, previously known as *Staphylococcus sciuri* [12,13], is emerging as a significant agent in gene resistance dissemination among Staphylococcaceae [14]. *M. sciuri* has garnered particular attention after several studies suggested that the *mecA*1 gene, ubiquitous in all methicillin-susceptible strains of *M. sciuri*, may be an evolutionary precursor to the homologous methicillin resistance gene *mecA* found in pathogenic MRSA strains. Of great concern, the resistance to methicillin in *M. sciuri* has been increasingly reported in recent years, particularly in Brazil [15]. Staphylococci and *M. sciuri* transmission among humans and animals can occur through direct contact or exposure to contaminated environments. These agents are a concern both for treating animal diseases and for potential public health consequences [16], making it increasingly necessary to expand studies in this area and control multidrug-resistant strains.

Bacterial antimicrobial resistance raises treatment costs, prolongs patient hospital stays, and can increase mortality rates. Additionally, the improper administration and use of antimicrobials compromise the patient’s clinical response, increase hospitalization costs, and can contribute to the emergence of multidrug-resistant bacteria [17]. The effort to discover and synthesize new drugs can take years; therefore, the rational use of antimicrobials is necessary to prevent resistance [18]. According to the World Health Organization (WHO), it is estimated that ten million deaths will occur annually by 2050 due to microbial resistance, and this impact will be significant on the global economy, approximately one hundred trillion dollars [19]. According to the WHO, the international phenomenon of antimicrobial resistance is as dangerous as a pandemic and threatens to destroy a century of medical progress [20].

Considering the potential exchange of different staphylococci species between humans and animals, this study aimed to identify and characterize samples of MRS and MRMs isolated from asymptomatic dogs and their owners in terms of species diversity and antimicrobial susceptibility patterns.

## 2. Results

### 2.1. Distribution of Staphylococci and M. sciuri Amongst Owners and Dogs

A total of 185 strains were recovered from dogs and owner samples, 127 isolates from asymptomatic dogs, and 58 from their respective owners. Of the strains isolated from dogs, 56.7% (72/127) were obtained from the nasal cavity, while 43.3% (55/127) were obtained from the perineum.

Regarding the species distribution, it was possible to confirm that of the 185 isolated strains, 22.1% (41/185) were identified as *S. epidermidis*, 18.9% (35/185) as *S. pseudintermedius*, 16.2% (30/185) as *M. sciuri*, 9.1% (17/185) as *S. aureus*, 8.1% (15/185) as *S. saprophyticus*, 7% (13/185) as *S. urealyticus*, 5.9% (11/185) as *S. hominis*, 3.2% (6/185) as *S. warneri* and *S. schleiferi* each, 2.7% (5/185) as *S. haemolyticus*, 1.6% (3/185) as *S. arlettae*, 1% (2/185) as *S. xylosus*, and 0.5% (1/185) as *S. intermedius*.

Considering the nasal isolates from dogs (72/127), the distribution of percentages by species was as follows: 22.2% (16/72) of *S. pseudintermedius*, 19.4% (14/72) of *M. sciuri*, 11.1% (8/72) of *S. saprophyticus*, 9.7% (7/72) of *S. epidermidis*, 8.3% (6/72) of *S. aureus*, 6.9% (5/72) of *S. urealyticus* and *S. hominis* each, 5.6% (4/72) of *S. warneri*, 4.2% (3/72) of *S. haemolyticus*, 2.8% (2/72) of *S. arlettae*, and 1.4% (1/72) of *S. schleiferi* and *S. xylosus* each. In the perineum (55/127), the distribution by species was as follows: 27.3% (15/55) of *S. pseudintermedius*, 20% (11/55) of *M. sciuri*, 10.9% (6/55) of *S. urealyticus*, 9.1% (5/55) of *S. epidermidis* and *S. schleiferi* each, 7.3% (4/55) of *S. hominis*, 5.4% (3/55) of *S. aureus*, 3.6% (2/55) of *S. saprophyticus*, and 1.8% (1/55) of *S. haemolyticus*, *S. warneri*, *S. arlettae*, and *S. xylosus* each.

A total of 58 strains were isolated from the nostrils of the owners. The distribution of percentages by species was as follows: 50% (29/58) of *S. epidermidis*, 13.8% (8/58) of *S. aureus*, 8.6% (5/58) of *M. sciuri* and *S. saprophyticus* each, 6.9% (4/58) of *S. pseudintermedius*, 3.5% (2/58) of *S. hominis* and *S. urealyticus* each, and 1.7% (1/58) of *S. haemolyticus*, *S. intermedius*, and *S. warneri*. No *S. schleiferi*, *S. arlettae*, or *S. xylosus* were isolated from the owners. All results of species distributions are depicted in Table 1.

### 2.2. Identification of MRS and MRMs

The *mecA* gene was detected in 33.5% (62/185) of the samples, comprising the majority—62.9% (39/62) from dogs and 37.1% (23/62) from their owners. When evaluating the percentage based on the number of dogs and owners, 43% (24/50) and 52.9% (18/34) carried microorganisms resistant to methicillin. Further considering all the *mecA* gene strains, the prevalence by species was 33.9% (21/62) in *S. epidermidis*, 16.1% (10/62) in *S. saprophyticus*, 14.5% in *M. sciuri* (9/62), 9.7% in *S. hominis* (6/62), 6.4% in *S. urealyticus* and *S. haemolyticus* each (4/62), 4.8% in *S. aureus* (3/62), 3.2% in *S. pseudintermedius* and *S. warneri* each (2/62), and 1.6% in *S. schleiferi* (1/62), as shown in Table 2.

In the dogs, among the isolates carrying the *mecA* gene, the prevalence by species was as follows: 20.5% (8/39) in *M. sciuri*, 17.9% (7/39) in *S. saprophyticus*, 15.4% (6/39) in *S. epidermidis*, 12.8% (5/39) in *S. hominis*, followed by 7.7% (3/39) in *S. urealyticus*, *S. haemolyticus*, and *S. aureus* each, 5.1% (2/39) in *S. warneri*, and finally, 2.6% (1/39) in *S. pseudintermedius* and *S. xylosus* each.

In the owners, the prevalence of the *mecA* gene by species was 65.2% (15/23) in *S. epidermidis*, 13% (3/23) in *S. saprophyticus*, followed by 4.4% (1/23) in *S. pseudintermedius*, *S. haemolyticus*, *S. hominis*, *S. urealyticus*, and *M. sciuri*.

### 2.3. Antimicrobial Resistance Pattern of MRS and MRMs

In the methicillin-resistant specimens isolated from the dogs, resistance to β -lactams was already expected. The results found were as follows: penicillin with 89.7% (35/39), cefoxitin with 46.1% (18/39) and oxacillin with 17.9% (7/39). Additionally, the highest percentage of resistance observed was to erythromycin (51.3%; 20/39), followed by trimethoprim–sulfamethoxazole (33.3%; 13/39). Considering the other antimicrobials, the following results were obtained: gentamicin (15.4%; 6/39) and clindamycin (12.8%; 5/39), with two isolates resistant to clindamycin due to erythromycin induction, forming a D zone; followed by ciprofloxacin, marbofloxacin, and enrofloxacin (10.3%; 4/39) each, and rifampicin (2.6%; 1/39). There were no resistant specimens to doxycycline.

In the methicillin-resistant specimens isolated from the owners, resistance to β-lactams was also already expected. The results found were as follows: penicillin with 82.6% (19/23), oxacillin with 69.6% (16/23), and cefoxitin with 21.7% (5/23). The other percentages of resistance found were to erythromycin (56.5%; 13/23), followed by clindamycin (26.1%; 6/23), with three isolates being resistant due to erythromycin induction, forming a D zone. Considering the other antimicrobials, the following results were obtained: marbofloxacin and ciprofloxacin (21.7%; 5/23) each, enrofloxacin and gentamicin (17.4%; 4/23) each, sulfamethoxazole–trimethoprim (8.7%; 2/23), and doxycycline (4.4%; 1/23). There were no specimens resistant to rifampicin, as demonstrated in Figure 1.

### 2.4. Comparative Study Between Dog–Owner Pairs MRS and MRMs

Fourteen pairs (dog–owner) sharing the same bacterial species were identified. Of these 14 pairs, 71% (10) had strains carrying the *mecA* gene in both hosts. The other four pairs presented the same bacterial species without the presence of the *mecA* gene, except for only one strain of *M. sciuri* isolated from a dog that was positive for the *mecA* gene (Table 3). In the antimicrobial susceptibility test, it can be observed that of the 14 pairs, 50% of them (7/14) presented the same antimicrobial susceptibility pattern. The rest showed different patterns, as shown in Table 3.

## 3. Discussion

In all 185 strains, the presence of the *mecA* gene, which confers resistance to β-lactam antimicrobial agents, was investigated. The gene was found in 33.5% of the isolates (62/185). When evaluating the percentage based on the number of dogs and their owners assessed, 43% (24/50) and 52.9% (18/34) carried microorganisms harboring the *mecA* gene, respectively. This gene can be transferred to other staphylococci occupying the same habitat [21].

Fourteen dog–owner pairs shared the same bacterial species carrying the *mecA* gene. Furthermore, seven of these pairs also exhibited the same antimicrobial resistance pattern, suggesting that there may have been transmission between the dog and its owner. In a study conducted by Gómez-Sans and collaborators [22], it was observed that nine out of ten dogs investigated were colonized by identical strains (characterized by the MLST technique) to those colonizing their owners, highlighting the sharing of this bacterium. Further analyses, such as the MLST technique, are necessary to confirm the sharing observed in this study.

No MRSA was simultaneously observed among the isolates from the dogs and their owners. This finding is consistent with the data reported by Suepaul and collaborators [23]. This may be because dogs are usually not colonized with *S. aureus*, although the temporary carriage of human *S. aureus* strains, especially for those of MRSA, is possible. Previous studies have shown that about 30% of healthy humans are colonized with *S. aureus*, while a third to half of humans are intermittently colonized, and the remainder are never colonized [23].

A methicillin-resistant *S. pseudintermedius* (MRSP) sample was found in a dog and its owner. Although *S. pseudintermedius* is not a common commensal in humans, this bacterium has been isolated from the nostrils of owners of dogs with staphylococcal pyoderma, with zoonotic origin confirmed through molecular testing [24]. This factor highlights the risk of transmitting *S. pseudintermedius*, including MRSP [25], especially for pet owners, due to the increasingly close contact between the hosts. As an aggravating factor, the antimicrobials used in the treatment of dogs are similar to those used in humans, and in most cases, the use is inappropriate [26]. According to Martins [27], the prevalence of MRSP has significantly increased and was more frequent in animals that had received previous antibiotic therapy.

Of the 12 antimicrobials tested in this study, we found resistance to 11, with strains demonstrating resistance to up to 10 different drugs. We also observed that MRS colonized seven dogs and their respective owners, and they showed identical resistance patterns, similar to the findings by Castro [28]. Only molecular analysis of these samples will allow for the identification of the sharing of the same MRS lineage among these hosts, indicating transmission of the bacterial species.

The canine samples demonstrated resistance to more antimicrobials when compared to human samples, which was also observed by Dotto [29]. This may have occurred because antimicrobial therapies in companion animals are often used without correct identification of the agent or without performing antimicrobial susceptibility tests. Additionally, easy access to veterinary medications increases the population’s inappropriate use of these products.

The increasing bacterial resistance to antimicrobials has become a global public health issue. It is important to emphasize that several studies support the evidence that methicillin resistance can be transmitted between microorganisms of animals and humans [30,31]. The role of companion animals as reservoirs of antimicrobial resistance, exacerbated by the horizontal transmission capability of resistance genes (*mecA*) among staphylococci species, should be researched in depth. This transmission is facilitated by close physical contact and by the fact that the antimicrobials used in small animal clinics are virtually the same as those used in human medicine. The quantification of this risk is highly problematic, as there are little data available on antimicrobial consumption in small animal clinics and antimicrobial susceptibility, as well as the prevalence of resistance genes among bacterial pathogens of pets [32].

In Brazil, antimicrobial drugs for veterinary use can still be purchased without a prescription. The frequent use of these antimicrobials may contribute to the population’s selective pressures on staphylococci. Methicillin and multidrug resistance are now common in both veterinary and human medicine, leading to findings that may foster future efforts to develop alternative approaches to control the staphylococcal infection burden.

Controlling AMR requires an integrated One Health approach that considers the interconnections between animals, humans, and environmental health. Control strategies, such as the rational use of antibiotics, the development of infection control programs in veterinary hospitals, and the continuing education of veterinarians, are essential to mitigate the spread of antimicrobial resistance [33]. We are aware that the present study still has some limitations, such as the reduced number of samples shared between dogs and their owners, as well as more in-depth analyses of this transmission, including data such as *SCCmec* type or clonality.

## 4. Material and Methods

### 4.1. Study Area, Population, and Samples

Samples were collected in veterinary clinics in Rio de Janeiro from May 2018 to March 2019. Sterile cotton swabs (Copan Diagnostic, Brescia, Italy) were used to collect samples from the nostrils and perineum of 50 dogs and the nostrils of their respective owners (34) during their clinic visits for routine check-ups or vaccinations—veterinarians collected the samples from the humans and animals. The samples were transported to the laboratory, where they were processed.

### 4.2. Isolation and Identification of Staphylococci and M. sciuri

Each swab was transferred to 3 mL of Tryptone Soy Broth (TSB) (Tryptone Soy Broth, Kasvi^®^, Pinhais, Italy) and incubated for 24 h at 37 °C. After incubation, the tubes were vortexed, and an entire loop of this culture was then plated on Mannitol Salt Agar (MSA) (Mannitol Salt Agar, Kasvi^®^, Italy) to isolate presumptive staphylococci and *M. sciuri*. The MSA plates were incubated for 24 h at 37 °C. After this initial incubation, three sample colonies with bacterial growth compatible with *staphylococci* and *M. sciuri* were selected for Gram staining and catalase testing. Gram-positive bacteria and those positive for the catalase test were stocked in a −20 °C freezer. All isolates were identified through Matrix-Assisted Laser Desorption Ionization Time-of-Flight Mass Spectrometry (MALDI-TOF Biotyper, Bruker, Billerica, MA, USA) following the manufacturer’s instructions, described as follows. A thin film of bacterial colony was applied in duplicate onto a 96-well steel plate (Bruker Daltonics, Billerica, MA, USA). Following drying at room temperature, 1 μL of 70% formic acid was applied to each colony and allowed to dry again at room temperature. Subsequently, 1 μL of matrix (a saturated solution of α-cyano-4-hydroxycinnamic acid [HCCA; Bruker Daltonics] in 50% acetonitrile and 2.5% trifluoroacetic acid) was applied to each colony and allowed to dry at room temperature before testing. The spectra were analyzed using a MALDI Biotyper automation control and the Bruker Biotyper 2.0 software and library (version 2.0, 3740 entries; Bruker Daltonics). Only results with scores ≥ 2000 were considered for this study.

### 4.3. Identification of the mecA Gene

The release of bacterial DNA from staphylococci and *M. sciuri* was performed using Chelex^®^ (Bio-Rad, Hercules, CA, USA) according to the methodology described by Walsh et al. [34]. Following DNA extraction, the samples were subjected to Polymerase Chain Reaction (PCR) for the detection of the *mecA* gene using the Promega Kit (Madison, WI, USA, GoTaq^®^ G2 DNA Polymerase) with the following concentrations of reagents: 5× Green Reaction Buffer, 0.2 mM dNTPs, 1.5 mM MgCl2, 10 pmol of forward primer (*mecA*F (5′TCCAGATTACAACTTCACCAGG3′)), 10 pmol of reverse primer (*mecA*R (5′CCACTTCATATCTTGTAACG3′)), 2 U of GoTaq, and 100 ng of DNA template in a total reaction volume of 25 μL. Amplification was carried out in a thermocycler (Applied Biosystems, Foster City, CA, USA), as described by Zhang et al. [35]. The reaction products were visualized through electrophoresis in 1% agarose gel in 0.5× TBE buffer (0.05 nM Tris, 1.25 mM EDTA, and 0.05 M boric acid). A 1 Kb molecular size marker (Invitrogen, Waltham, MA, USA) was used. The expected product size was 162 bp. The *S. aureus* USA100 strain was used as positive control.

### 4.4. Antimicrobial Susceptibility Testing of MRS and MRM Strains

Samples that carried the *mecA* gene were analyzed for resistance to other antimicrobial agents using the disk diffusion method (Kirby–Bauer) on Mueller–Hinton agar (Becton, Dickinson and Company BD, Franklin Lakes, NJ, USA), according to the recommendations of the Clinical and Laboratory Standards Institute [36,37]. After bacterial growth on TSA medium at 37 °C for 24 h, a bacterial suspension was prepared in sterile 0.85% saline, achieving a turbidity of 0.5 on the McFarland scale. Subsequently, using a sterile swab, the suspension was inoculated onto a plate containing Mueller–Hinton agar. Twelve antimicrobial discs were used, belonging to eight classes: beta-lactams (cefoxitin 30 µg, oxacillin 1 µg, penicillin G 10 µg), fluoroquinolones (ciprofloxacin 5 µg, enrofloxacin 5 µg, marbofloxacin 5 µg), lincosamide (clindamycin 2 µg), macrolide (erythromycin 15 µg), aminoglycoside (gentamicin 10 µg), rifampicin 5 µg, tetracycline (doxycycline 30 µg), and sulfonamide–trimethoprim (cotrimoxazole 25 µg) (CECON, Vila Sonia, São Paulo, Brazil). The plates were incubated aerobically at 37 °C for 24 h, and the diameters of the zones were measured manually using an Antibiotic Zone Scale-C™ (HiMedia, Thane, India). The *S. aureus* ATCC 25923 strain was used as positive control.

## 5. Conclusions

Several species of staphylococci were isolated from dogs and their owners. The most prevalent were *S. pseudintermedius* in the dogs and *S. epidermidis* in the owners. MRS and MRMs strains were isolated from both the dogs and their owners. The most prevalent methicillin-resistant strain in the dogs was *M. sciuri*, while in the owners, it was *S. epidermidis*. The MRS strains from the dogs exhibited high resistance rates to other antimicrobials such as erythromycin and sulfamethoxazole–trimethoprim. Meanwhile, the MRS strains from the owners showed interesting resistance rates to erythromycin and clindamycin.

## Figures and Tables

**Figure 1 antibiotics-14-00409-f001:**
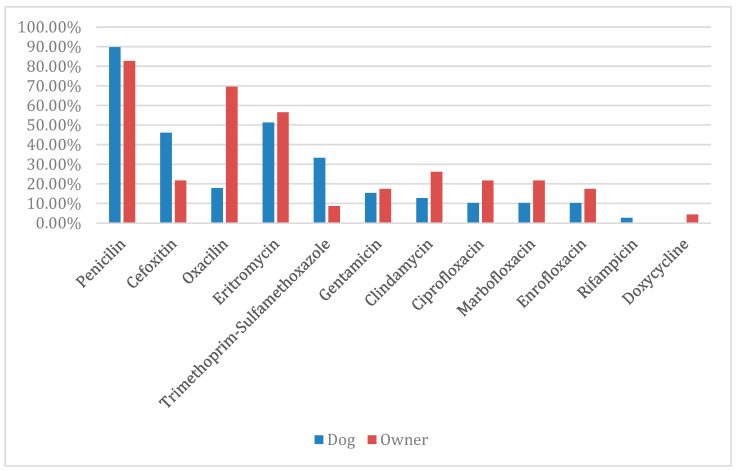
Percentage of antimicrobial resistance pattern of MRS and MRMs.

**Table 1 antibiotics-14-00409-t001:** Distribution of staphylococci and *M. sciuri* isolated from dogs and their owners (N: number).

Species	Isolated from Dogs % (N)		Isolated from Owners % (N)	TOTAL % (N)
	Nostrils	Perineum	Nostrils	General
*S. pseudintermedius*	22.2 (16)	27.3 (15)	6.9 (4)	18.9 (35)
*M. sciuri*	19.4 (14)	20 (11)	8.6 (5)	16.2 (30)
*S. epidermidis*	9.7 (7)	9.1 (5)	50 (29)	22.1 (41)
*S. urealyticus*	6.9 (5)	10.9 (6)	3.5 (2)	7 (13)
*S. saprophyticus*	11.1 (8)	3.6 (2)	8.6 (5)	8.1 (15)
*S. aureus*	8.3 (6)	5.4 (3)	13.8 (8)	9.1 (17)
*S. hominis*	6.9 (5)	7.3 (4)	3.5 (2)	5.9 (11)
*S. schleiferi*	1.4 (1)	9.1 (5)	0	3.2 (6)
*S. warneri*	5.6 (4)	1.8 (1)	1.7 (1)	3.2 (6)
*S. haemolyticus*	4.2 (3)	1.8 (1)	1.7 (1)	2.7 (5)
*S. arlettae*	2.8 (2)	1.8 (1)	0	1.6 (3)
*S. xylosus*	1.4 (1)	1.8 (1)	0	1 (2)
*S. intermedius*	0	0	1.7 (1)	0.5 (1)
Total % (N)	56.7 (72)	43.3 (55)	100 (58)	100 (185)

**Table 2 antibiotics-14-00409-t002:** Prevalence of *mecA* gene.

Species	*mecA* Gene (%)
*S. epidermidis*	33.9
*S, saprophyticus*	16.1
*M. sciuri*	14.5
*S. hominis*	9.7
*S. urealyticus*	6.4
*S. haemolyticus*	6.4
*S. aureus*	4.8
*S. pseudintermedius*	3.2
*S. warneri*	3.2
*S. schleiferi*	1.6

**Table 3 antibiotics-14-00409-t003:** Demonstration of the 14 dog–owner pairs.

Pairs	Sample Identification	Bacterial Species	*mecA* Gene	Site	CFO	PEN	OXA	GEN	CIP	ENO	MBF	ST	ERI	CLI	RIF	DOX	Multidrug-Resistant
1	7 (D)	*S. warneri*	+	N	34	18	----	28	30	30	32	22	R	R	42+	36	*
	333 (O)	*S. warneri*	+	N	24	16	----	24	28	32	30	26	R	R	38	14	*
2	33.1 (D)	*S. epidermidis*	+	N	----	18	12	30	34	34	30	22	12	30	42	24	
	340 (O)	*S. epidermidis*	+	N	----	34	12	24	32	30	30	20	10	28	40	30	
3	83.2 (D)	*S. epidermidis*	+	P	----	14	14	30	34	36	34	28	28	28	42	30	
	353 (O)	*S. epidermidis*	+	N	----	10	14	24	36	32	34	26	30	30	42	34	
4	107.1 (D)	*S. saprophyticus*	+	P	26	14	----	30	32	34	30	36	30	26	38	34	
	358.2 (O)	*S. saprophyticus*	+	N	18	10	----	32	32	36	30	34	30	26	40	38	
5	316.2 (D)	*S. epidermidis*	+	N	----	36	14	30	36	36	36	24	34	34	42+	30	
	410 (O)	*S. epidermidis*	+	N	----	36	18	26	34	36	36	24	32	32	42	30	
6	99 (D)	*S. hominis*	+	N	28	R	----	8	R	R	12	R	12	36	42	36	*
	347 (O)	*S. hominis*	+	N	22	R	----	10	R	R	10	R	R	28	42	32	*
7	144 (D)	*S. pseudintermedius*	+	N	----	10	16	12	R	R	8	R	10	18	42	36	*
	367 (O)	*S. pseudintermedius*	+	N	----	10	24	R	R	R	R	R	R	16	36	34	*
8	23 (D)	*S. haemolyticus*	+	N	10	R	----	R	R	R	R	R	R	R	42+	28	*
	338 (O)	*S. haemolyticus*	+	N	12	12	----	R	26	24	30	24	R	20	40	22	*
9	216 (D)	*S. epidermidis*	+	N	----	16	12	16	30	36	34	24	28	30	12	22	
	387 (O)	*S. epidermidis*	+	N	----	16	14	26	30	30	28	20	10	28	42	32	
10	223.2 (D)	*S. saprophyticus*	+	P	16	10	----	28	30	28	26	30	28	30	38	34	
	388 (O)	*S. saprophyticus*	+	N	20	12	----	R	32	34	32	28	R	24 (D)	42	16	*
11	173 (D)	*S. aureus*	-	P	34	20	----	14	26	30	26	32	6	26	36	34	
	365 (O)	*S. aureus*	-	N	36	18	----	12	28	30	26	30	6	26	36	30	*
12	319 (D)	*S. aureus*	-	N	30	18	----	22	28	26	26	30	6	26	34	34	
	412 (O)	*S. aureus*	-	N	34	16	----	22	22	28	28	30	R	26 (D)	32	30	*
13	420 (D)	*S. aureus*	-	N	32	14	----	10	28	24	28	30	6	26 (D)	38	30	*
	416 (O)	*S. aureus*	-	N	36	18	----	24	26	30	30	32	R	30 (D)	38	30	*
14	126 (D)	*M. sciuri*	-	P	26	28	----	26	26	26	26	26	26	18	32	30	
	130 (D)	*M. sciuri*	+	N	26	28	----	26	26	26	24	26	26	20	32	30	
	356 (O)	*M. sciuri*	-	N	26	30	----	24	24	24	24	26	26	18	30	28	

(D—dog; O—owner; N—nostril; P—perineum; CFO—cefoxitin; PEN—penicillin; OXA—oxacillin; GEN—gentamicin; CIP—ciprofloxacin; ENO—enrofloxacin; MBF—marbofloxacin; ST—sulfamethoxazole–trimethoprim; ERI—erythromycin; CLI—clindamycin; RIF—rifampicin; DOX—doxycycline; “D” in red—isolates that showed D zone; *—multidrug-resistant samples).

## Data Availability

The data are contained within the article.

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
