# Peer review of "Frequency, Distribution, and Antimicrobial Resistance of Methicillin-Resistant Staphylococci and Mammaliicoccus sciuri Isolated from Dogs and Their Owners in Rio de Janeiro"

_antibiotics, 2025, doi:10.3390/antibiotics14040409_

Round 1
Reviewer 1 Report
Comments and Suggestions for Authors
The manuscript entitled: “Frequency, distribution, and antimicrobial resistance of methi- cillin-resistant staphylococci and Mammaliicoccus sciuri isolated from dogs and their owners in Rio de Janeiro” is a well designed study. Here the authors have detected the presence of met. resistant Staph and Mamma. spp. from dog and dog owners. The findings shows the risk of pet owners in picking up these resistant isolates from dogs.
Comments:
Please make the introduction a bit short.
Please make a graph/figure using some data from table 1
Show the PCR picture of mecA gene detection and plates of sensitivity test. What were the PCR control and sensitivity test assay control??
Antibiotic susceptibility testing: The methodology follows CLSI guidelines, but the minimum inhibitory concentration (MIC) testing would provide a stronger and more clinically relevant analysis.
The discussion is well-written but lacks comparative analysis with similar studies outside Brazil. How do these findings compare with AMR E. faecalis reports from other countries?
Write on limitation of the study…few samples??
Line 294: MRS strains from dogs exhibited significant resistance rates to other antibiotocs…….you must SHOW the evidence..statistical analysis …pearson coorelation or soe analysis please...no statistical analysis in methodology??
to other antimicrobials, such
In this connection One Health recommendations could be mentiojned. What interventions should be prioritized to mitigate the spread of these pathigens? Are there policy recommendations for antibiotic use in dogs?
The manuscript is mostly well-written but has grammatical errors and awkward phrasing in some sections. A native English speaker or professional editing service should refine the language.
Comments on the Quality of English Language
The manuscript entitled: “Frequency, distribution, and antimicrobial resistance of methi- cillin-resistant staphylococci and Mammaliicoccus sciuri isolated from dogs and their owners in Rio de Janeiro” is a well designed study. Here the authors have detected the presence of met. resistant Staph and Mamma. spp. from dog and dog owners. The findings shows the risk of pet owners in picking up these resistant isolates from dogs.
Comments:
Please make the introduction a bit short.
Please make a graph/figure using some data from table 1
Show the PCR picture of mecA gene detection and plates of sensitivity test. What were the PCR control and sensitivity test assay control??
Antibiotic susceptibility testing: The methodology follows CLSI guidelines, but the minimum inhibitory concentration (MIC) testing would provide a stronger and more clinically relevant analysis.
The discussion is well-written but lacks comparative analysis with similar studies outside Brazil. How do these findings compare with AMR E. faecalis reports from other countries?
Write on the limitation of the study…few samples??
Line
Line 294: MRS strains from dogs exhibited significant resistance rates to other antibiotocs…….you must SHOW the evidence..statistical analysis …pearson coorelation or soe analysis please...no statistical analysis in methodology??
to other antimicrobials, such
In this connection One Health recommendations could be mentiojned. What interventions should be prioritized to mitigate the spread of these pathigens? Are there policy recommendations for antibiotic use in dogs?
The manuscript is mostly well-written but has grammatical errors and awkward phrasing in some sections. A native English speaker or professional editing service should refine the language.
Author Response
REVIEWER 1:
Comments:
- Please make the introduction a bit short.
Introduction was made shorter.
- Please make a graph/figure using some data from table 1.
Thank you for the comment. But we chose to present the data in table format, as it is easier to see all the information.
- Show the PCR picture of mecA gene detection and plates of sensitivity test. What were the PCR control and sensitivity test assay control??
PCR control was added. (Now it is in line 301 and line 318).
- Antibiotic susceptibility testing: The methodology follows CLSI guidelines, but the minimum inhibitory concentration (MIC) testing would provide a stronger and more
clinically relevant analysis.
Thank you for the comment. We chose to perform only disc diffusion since it is reliable and also recommended by CLSI for Staphylococcus.
- The discussion is well-written but lacks comparative analysis with similar studies outside Brazil. How do these findings compare with AMR E. faecalis reports from other countries?
We chose to work only with Staphylococcus sp. and M. sciuri. Therefore, discussion follows with reports from these species.
- Write on limitation of the study…few samples??
Limitations of the study were added (It is in line 256).
- Line 294: MRS strains from dogs exhibited significant resistance rates to other antibiotocs…….you must SHOW the evidence..statistical analysis …pearson coorelation or soe analysis please...no statistical analysis in methodology??
The phrase was corrected. No significant difference was observed. We corrected this (It is in line 325).
- In this connection One Health recommendations could be mentioned. What interventions should be prioritized to mitigate the spread of these pathigens? Are there policy recommendations for antibiotic use in dogs?
Thank you for the comment. Suggestion accepted. (Now it is in line 252).
9. The manuscript is mostly well-written but has grammatical errors and awkward phrasing in some sections. A native English speaker or professional editing service should refine the language.
Thank you for the comment. An English review was conducted in the manuscript.
Reviewer 2 Report
Comments and Suggestions for Authors
The manuscript by Fernanda Cruz Bonnard el al has identified and characterized antimicrobial resistance MRSA and MRMs isolated from dogs and their owners in Rio de Janeiro. The finding provides crucial data for local and international reference. The manuscript is well-written and interesting. However, there are some issues we spotted in the manuscript that need to be addressed before considering this manuscript for publication.
Comments to Authors
- The organism’s name must be correct and italicized throughout the manuscript.
- Please be consistent when writing the organism's name eg: Staphylococcus aureus at the first time mentioned then S. aureus throughout the paper. Please check it all
- There is no mention of QC strains used side-by-side with the test when performing the AST test which leads to the assumption that the testing was done without quality controls
- Line 159: Please replace the word “Graphic 1” to Figure 1
- Line 159: why the author did not include the percentage of resistance of the Cefoxitin , Oxacillin , and Penicillin G and incorporate them in the figure? Is there any reason for that? I have noted that 12 antimicrobial agents were tested but only 9 were mentioned in the result.
- Line 279: Human [35] and Animal [36]. This is not clear. Please rewrite the statement
Author Response
REVIEWER 2:
Comments to Authors:
- The organism’s name must be correct and italicized throughout the manuscript.
All organisms name were italicized.
- Please be consistent when writing the organism’s name eg: Staphylococcus aureus at the first time mentioned then S. aureus throughout the paper. Please check it all.
The manuscript was checked and corrections were made.
- There is no mention of QC strains used side-by-side with the test when performing the AST test which leads to the assumption that the testing was done without quality controls.
Quality control was added in the material and methods (line 318).
- Line 159: Please replace the word “Graphic 1” to Figure 1.
The word was replaced (Now it is in line 186).
- Line 159: why the author did not include the percentage of resistance of the Cefoxitin, Oxacillin, and Penicillin G and incorporate them in the figure? Is there any reason for that? I have noted that 12 antimicrobial agents were tested but only 9 were mentioned in the result.
The other antibiotics were included in the results (Now it is in line 157).
- Line 279: Human [35] and Animal [36]. This is not clear. Please rewrite the statement
The statement was rewrote (Now it is in line 307).
Reviewer 3 Report
Comments and Suggestions for Authors
The manuscript titled “Frequency, distribution, and antimicrobial resistance of methicillin-resistant staphylococci and Mammaliicoccus sciuri isolated from dogs and their owners in Rio de Janeiro” explores the prevalence of methicillin-resistant staphylococci and Mammaliicoccus sciuri in dogs and their owners in Rio de Janeiro.
Antibiotic resistance poses a significant public health challenge, and this study provides valuable insights into the potential role of dogs as carriers of antibiotic-resistant strains transmissible to humans. The manuscript is well-written, with the authors presenting their findings clearly and in a well-structured manner.
Please consider these suggestions for improving this manuscript:
Minor
Abstract
Line 9. “Staphylococcus” instead of “Staphylococcus”. It should be in Italics
Line 12. Staphylococci shouldn’t be in Italics
Discussion
Line 192. I suggest authors also add [23] after “Suepaul et al.”
Line 233. Delete “.” after staphylococci
Material and Methods
4.3 Identification of mecA gene.
- Authors should provide details on the kit they used for the PCR
- What was the size (length) of the amplicon?
Line 284. Authors should provide more details on the antibiotic discs (company etc)
Author Response
REVIEWER 3:
Minor
Abstract
- Line 9. “Staphylococcus” instead of “Staphylococcus”. It should be in Italics.
Italics were added in all the text (Now it is in line 12).
- Line 12. Staphylococci shouldn’t be in Italics
The line was corrected (Now it is in line 15).
Discussion
- Line 192. I suggest authors also add [23] after “Suepaul et al.”
We accepted the suggestion (Now it is in line 206).
- Line 233. Delete “.” after staphylococci
We accepted the suggestion (Now it is in line 248).
Material and Methods
4.3 Identification of mecA gene.
- Authors should provide details on the kit they used for the PCR
Details on the kit were provided (Now it is in line 292).
- What was the size (length) of the amplicon?
Size of the amplicon was added (Now it is in line 301).
- Line 284. Authors should provide more details on the antibiotic discs (company etc)
More details on antibiotics disc were provided (Now it is in line 316).